# Future Antimicrobials: Natural and Functionalized Phenolics

**DOI:** 10.3390/molecules28031114

**Published:** 2023-01-22

**Authors:** Andrei Lobiuc, Naomi-Eunicia Pavăl, Ionel I. Mangalagiu, Roxana Gheorghiță, Gabriel-Ciprian Teliban, Dorina Amăriucăi-Mantu, Vasile Stoleru

**Affiliations:** 1Faculty of Medicine and Biological Sciences, “Ştefan cel Mare” University, 720229 Suceava, Romania; 2Faculty of Chemistry, “Alexandru Ioan Cuza” University, 700506 Iasi, Romania; 3Department of Horticulture Technologies, “Ion Ionescu de la Brad” University of Life Sciences, 700490 Iasi, Romania

**Keywords:** flavonoids, tannins, lignans, minimal inhibitory concentration, esterification, alkylation

## Abstract

With incidence of antimicrobial resistance rising globally, there is a continuous need for development of new antimicrobial molecules. Phenolic compounds having a versatile scaffold that allows for a broad range of chemical additions; they also exhibit potent antimicrobial activities which can be enhanced significantly through functionalization. Synthetic routes such as esterification, phosphorylation, hydroxylation or enzymatic conjugation may increase the antimicrobial activity of compounds and reduce minimal concentrations needed. With potent action mechanisms interfering with bacterial cell wall synthesis, DNA replication or enzyme production, phenolics can target multiple sites in bacteria, leading to a much higher sensitivity of cells towards these natural compounds. The current review summarizes some of the most important knowledge on functionalization of natural phenolic compounds and the effects on their antimicrobial activity.

## 1. Introduction

Polyphenols comprise a variety of biologically active compounds that are widely distributed in plants and represent the most common phytochemicals with beneficial health effects. Apart from their role in plant physiology (i.e., pigmentation, growth, reproduction and resistance to pathogens), phenolic compounds are part of human diet and have beneficial implications in human health [1]. Phenolics have specific chemical reactivity which cause significant biological activities, ranging from antioxidants to antiproliferative compounds. Furthermore, these activities are profoundly related to chemical groups grafted onto the phenolic core, which broadens the diversity and intensity of the phenolics’ biological activities while also offering opportunities for synthesizing new compounds. For instance, the radical-scavenging and metal-chelating activities are structure dependent, as the number of hydroxyl groups and their position related to carboxyl group influences the antioxidant activity [2]. Antioxidant mechanisms, such as hydrogen atom transfer (HAT) and proton-coupled electron transfer (PCET), are correlated with thermodynamic properties of the radical. The arrangement of hydroxyl groups in the ring structure influences the stability of radicals which are formed by the loss of H to the free radicals. Electromeric effects mean that the ortho position of hydroxyl groups confers high stability to the radicals. Regarding hydroxyl groups, the higher the number, the stronger antioxidant and other properties [3,4]. Considering antibacterial activity, plant polyphenols can act against bacterial cells through several mechanisms, such as interaction with proteins and bacterial cell walls, alteration of the cytoplasmic functions and of membrane permeability, inhibition of energy metabolism and DNA damage or inhibition of nucleic acids synthesis by bacteria cells. At the DNA level, the planarity of the molecule and the hydrophobic core mean that polyphenols can penetrate the DNA helix during the replication, recombination, repair and transcription mechanisms. Moreover, hydroxyl groups of phenolic compounds allow the molecules to form hydrogen bonds with the nucleic acid bases. Phenolics also interact with synthetic pathways; for instance, polyphenols inhibit topoisomerase or DNA gyrase activity. Polyphenols can also combine with metals such as Cu^2+^ and form complexes that modify DNA stability. The mechanism of inhibition differs, depending on the structure of the polyphenols as well as the bacterial species. The favorable character of the molecule, hydrophilic or hydrophobic, depends on the action sites, and thus the amphipathic character of phenolic compounds plays a very important role in antibacterial activity [5].

### 1.1. Classification of Major Classes of Antibacterial Phenolics

Chemically, polyphenols, a class with a rich diversity of compounds, range from simple molecules, with an aromatic ring and one or more hydroxyl groups, to highly polymerized compounds (Table 1). Polyphenols can also be classified by their source of origin, natural distribution and biological activity. Based on the number of aromatic rings and the structural elements, polyphenols are classified as phenolic acids, flavonoids, lignans, stilbenes and tannins [6]. Given the high structural diversity, polyphenols act on different action sites at the bacterial cell level, and their action depends on the specific properties given by functional groups and aromatic rings. At the membrane level, the activity of plant polyphenols is favored by the lipophilic character of molecules, and thus the phenolic compound can modify membrane permeability and bind to enzymes, which leads to changes in intracellular functions [7]. Plant polyphenols with pyrogallol groups have a more significant activity than polyphenols with catechol or resorcinol groups; as such, pyrogallol group presence is an indicator for potent antibacterial activity [8]. In the metabolic pathway of bacterial cells, functional groups can inhibit or augment enzymatic activity. For example, the inhibitory effect of polyphenols against α-amylase activity is influenced by the hydroxylation degree, the presence of an unsaturated bond, and the glycosylation, methylation and methoxylation degrees of the molecule [9].

### 1.2. Biosynthesis, Natural and Synthetic Sources

Polyphenols are widespread in plants [15]; one of their abilities is to ensure protection against oxidative stress through the ability to both scavenge free radicals and chelation of transition metals and also inhibit enzymes such as xanthine oxidase, lipoxygenase, protein kinase C, cyclooxygenase, microsomal monooxygenase, mitochondrial succinoxidase and NADPH oxidase, that generate free radicals [16]. Their concentration depends on the role they perform in plants and is influenced by different conditions of plant growth, preparation or storage. The majority of the phenolic compounds appear as glycosylated, esterified or condensed forms, with some compounds occurring as free forms [17].

Polyphenols occur through two metabolic pathways: shikimate pathway and malonic acid pathway. The malonic acid pathway is less significant in plants but is an important biosynthesis pathway for phenolics in bacteria and fungi [18]. In plants, most polyphenols are synthesized by the shikimate pathway. Polymeric polyphenols are synthesized from monomeric compounds. For example, hydroxycinnamic acids are building blocks for lignans and hydrolysable tannins are synthesized from derivates of benzoic acid. Different factors can influence the production rate of polyphenolics in plants through activation and inhibition of enzymes. Their synthesis can be regulated by several environmental factors such as temperature, soil properties, light irradiation, irrigation and others [19]. Their wide distribution in plants means that phenolic compounds are present in the human diet, with high concentrations in beverages and food of plant origin.

## 2. Antibacterial Activity

The structural diversity of polyphenols incurs activity against the bacterial cell, altering cell structure and morphology or producing imbalances in bacterial metabolism. Due to hydroxylic groups, phenolic compounds can interact with diverse sites of the bacterial cell.

### 2.1. Antibacterial Activity of Phenolic Acids

Phenolic acids have antimicrobial activity against both Gram-positive and Gram-negative bacteria. Their activity is weaker than other phenolic compounds, and their antimicrobial potential depends on species, strains and the chemical structure of compounds, in particular the number and position of substituents in the benzene ring and the saturated chain length [20,21]. For instance, one proposed mechanism of action of phenolic acids is a decrease in extracellular pH, as is the case with gallic and chlorogenic acids [22]. Other than its intrinsic chemical properties, the antibacterial activity depends on the interaction site with target molecules, inducing different action mechanisms. Phenolic acids have membrane-active properties against bacteria, which cause leakage of cell constituents, including nucleic acids, proteins and inorganic ions such as potassium or phosphate. They act both at the membrane and cytoplasmic levels. 

Another mechanism of action at the membrane level is hyperacidification, which alters cell membrane potential. The dissociation of phenolic acids makes the membrane more permeable, causing the sodium-potassium pump to be affected. Gram-positive bacteria are more susceptible to this antibacterial mechanism because of the absence of an outer membrane. It was found that a high concentration of phenolic acids (1000 µg/mL) has antibacterial activity against *lactobacilli*, such as *L. paraplantarum* LCH7, *L. plantarum* LCH17, *L. fermentum* LPH1, *L. fermentum* CECT 5716, *L. brevis* LCH23 and *L. coryniformis* CECT 5711 [23]. Phenolic acids such as gallic acid may also change the charge, hydrophobicity and permeability of the membrane or, in the case of Gram-negative bacteria, disintegrate it through divalent cations chelation [24]. Meanwhile, the antibacterial activity depends on the strains as well as the number and position of substituents in the benzene ring because of the influence of the dissociation capacity; for example, phenolic acids dissociate at the cell membrane level of Gram-positive pathogens and inhibit the growth of bacteria such as *S. aureus* EP167 and *S. aureus* ATCC 29213, with an IC of 1000 µg/mL and 667 µg/mL respectively [23].

In the case of Gram-negative bacteria, phenolic acids act at the cytoplasmic level. The antimicrobial activity depends on the concentration of the phenolic acids in their undissociated form. In this form, their partially lipophilic nature allows phenolic acids to cross the cell membrane through passive diffusion. Acidification of the phenolic acids is intracellular, decreasing pH-value in the cytoplasm. By decreasing the pH-value and disrupting the cell membrane structure, they cause protein denaturation. The membrane damage leads to an increase in permeability, and thus an increase in potassium effluxes. Antibacterial activity of hydroxycinnamic acids (ferulic acid) and hydroxybenzoic acids (gallic acid) was tested on the cell membrane of *E. coli, P. aeruginosa, S. aureus* and *L. monocytogenes*, and it was found that ferulic acid acts against bacteria with MIC between 100 and 1250 µg/mL, and that gallic acid acts with MIC in the range 500–2000 µg/mL [25]. The inhibitory effect on the bacteria growth depends on the side chain, because lipophilicity increases with their length, and the pH-value. In the case of *L. monocytogenes*, cinnamic acid has increased antibacterial effect at decreased pH, with MIC values of 6.09 mM [26].

Phenolic acids can also act against bacteria through non-membrane mechanisms. For example, *p*-coumaric acid can bind bacterial DNA. The action of *p*-coumaric acid on *Shigella* DNA leads to an increased molecular planarity of DNA, which demonstrates the localization of planar aromatic acid molecules between base pairs of DNA in a hydrophobic environment [27].

### 2.2. Antibacterial Activity of Flavonoids

Owing to their chemical structure, flavonoids have multiple bacteria cellular targets. For instance, flavonoids have antibacterial activity by interacting with cell membranes. The interaction between flavonoids and the lipid bilayer is influenced by the number and distribution of the hydroxyl groups, the polymerization degree and the presence of a methoxy group in the C ring. Hydrophobic flavonoids can penetrate the nonpolar core of the bacteria cell membrane. Another possible interaction is associated with the formation of hydrogen bonds between hydrophilic flavonoids and the polar headgroups of lipids [28]. Naringenin and sophoraflavanone G are two flavonoids with antibacterial activity against MRSA and *streptococci*. These flavonoids reduce the fluidity of outer and inner layers of membranes of bacterial cells [29].

Catechins are a group of flavonoids with significant activity against both Gram-positive and Gram-negative bacteria. The flavonoids can penetrate the lipid bilayer and thus disrupt the barrier function of cell membrane or can cause membrane fusion, a process that results in leakage of intramembranous materials [30]. Other flavonoids have antibacterial activity against *S. aureus* ATCC 700699, *E. coli* 0157:H7, *V. parahemoliticus* ATCC 17082, *Y. enterocolitica* ATCC 23715 at a MIC value of 15.63 µg/mL [31], *S. maltophilia* with a very low MIC value (4 µg/mL) [32], *S. epidermidis* and *Enterococcus faecalis* [33].

Along with the action at membrane level, flavonoids inhibit nucleic acids synthesis. The inhibitory action on DNA synthesis can be explained by the important role of the B ring of flavonoids, which intercalate between nucleic acid bases or form H bonds. This mechanism occurs through the inhibition of DNA gyrase and depends on the hydroxylation of the B ring. It was found that many flavonoids of varying structure have weak inhibition activity against DNA gyrase and against *S. epidermidis*, *S. aureus*, *E. coli* and *S. maltophilia* with MICs more than 250 µg/mL [34]. Numerous steps in nucleic acids synthesis involve enzymes, which are inhibited by flavonoid compounds. For example, both baicalein, a flavonoid from *S. baicalensis*, and quercetin inhibit bacterial invasion of epithelial cells by deactivating the type III secretion system. The interaction between these flavonoids and type III secretion system decreases the bacterial virulence of *S. enterica*, but without affecting bacterial growth [35].

Flavones, a subgroup of flavonoids, are inhibitors of helicases, which play an important role in acid nucleic metabolism. Thus, flavones can inhibit growth of pathogenic bacteria, such as *K. pneumoniae* and *E.coli*. Myricetin, for instance, inhibits replicative hexameric DNA helicase and a variety of helicases with different function and origin [36].

Quercetin, apigenin and sakuranetin are three flavonoids with competitive inhibitory effect against β-hydroxyacyl-acyl carrier protein dehydratase from *H. pylori*. Their mechanism involves the binding to the substrate, which prevents the substrate from accessing the active site. These flavonoids also inhibit β hydroxyacyl-ACP dehydrase from *H. pylori* with IC_50_ value 39.3 ± 2.7 μM, 11.0 ± 2.5 μM and 2.0 ± 0.1 μM, respectively. Quercetin also induces DNA cleavage and inhibits bacterial gyrase [37]. Such enzymes are involved in the type II fatty acid biosynthesis pathway and alteration of their activity may, thus, interact with membrane lipid homeostasis in the growth and stationary phases as well as in physical and nutrient states [38].

The antibacterial activity of flavonoids includes the prevention or inhibition of biofilm formation. Single cells attach randomly to a surface and form three-dimensional biofilms by dividing and developing into mature cells. Initially, flavonoids cause aggregations of bacterial cells. Multicellular aggregates of *S. aureus* were observed after incubation with two flavonoids: epigallocatechin-3-gallate and 3-O-octanoyl-epicatechin. Flavonoids cause bacterial aggregation through the membrane fusion, but then reduce the active nutrient uptake, meaning that bacteria cannot survive. Flavonoids also exhibit antibiofilm activity; for example, epicatechin inhibits biofilm formation of *S. aureus* ATCC 29213 and *S. mutans* but does not reduce *L. monocytogenes* biofilms. Apigenin-6-C-glycoside and 5,7,4′-trihydroxyflavanol have antibiofilm activity against *S. aureus* ATCC 29213 [33]. Some flavonoids inhibit biofilm formation by Gram-positive bacteria through interactions sortases enzymes found in the cytoplasmic membrane bacteria that catalyze the assembly of cells and are important for establishment of infection [11]. 

### 2.3. Antibacterial Activity of Lignans 

Some lignan compounds have activity against diverse bacterial species and act mainly at the cell membrane level. For example, (−)-nortrachelogenin (5 µg/mL) exhibits an inhibitory effect against bacteria through several modes. In one mode, the compound acts at the cytoplasmic membrane level, causing membrane permeabilization as well as damage to the membrane. Against *E. coli* O157 cells, it was observed that (−)-nortrachelogenin causes depolarization to the cytoplasmic membrane and damage of the membrane. These alterations of the cell membrane affect intracellular environment by modifying the pH and ion gradient. The treatment of *E. Coli* O157 with lignan leads to leakage of intracellular potassium ion, and thus produces a negative bacterial membrane potential [39]. Pinoresinol from *Cinnamomum camphora* leaves (MIC in the range 3.9–31.25 µg/mL) damages the cell membrane of *P. aeruginosa* and *B. subtilis*, leading to bacteria cell death [40]; lignan 3′-demethoxy-6-O-demethylisoguaiacin has antibacterial activity against *S. aureus* (MIC 25 µg/mL), *E. faecalis* (MIC 12.5 µg/mL), *E. coli* (MIC 50 µg/mL) and *E. cloacae* (MIC 12.5 µg/mL). Dihydroguaiaretic acid has activity against *S. aureus* (MIC 50 µg/mL) and multidrug-resistant strains of *Mycobacterium tuberculosis* (MIC 12, 5–50 µg/mL), and 4-epi-larreatricin has activity against *E. cloacae* (MIC 12.5 µg/mL) and multidrug-resistant strains of *M. tuberculosis* (MIC 25 µg/mL) [41].

### 2.4. Antibacterial Activity of Stilbenes

Different antibacterial mechanisms can be observed among the stilbenes. Resveratrol and other stilbenes act on diverse bacterial targets, including both Gram-positive and Gram-negative bacteria. Stilbenes are bacteriostatic, possessing an inhibitory effect on the growth of bacteria, with limited bactericidal activity. Stilbenes gradually disrupt the cellular functions and cause cell death in several days. The activity of resveratrol against *P. acnes* growth involves the loss of well-defined extracellular fimbrial structures and an intramembranous edema, which causes a loss of membrane definition [42]. Stilbenes act co-operatively with antibiotics against bacteria, interfering with enzymes that rewind DNA after being copied and thus stop DNA synthesis [13].

Stilbenes inhibit the phosphotransferase system, a widespread system in bacteria that catalyzes phosphorylation and carbohydrate transport. They act on several proteins that are carbohydrate specific type II transporters, including specific transporters of mannose and sorbitol. The action of stilbenes on the phosphotransferase system significantly alters the energy metabolism pathway of *Enterococcus* strains, with MIC values between 0.5 and 2 µg/mL [43].

### 2.5. Antibacterial Activity of Tannins

Tannins have an inhibitory effect against pathogenic bacteria through different mechanisms of action. Iron chelation is one action modes against bacteria, as iron is essential for bacterial growth. Bacteria produces siderophores that bind iron and form essential compounds for optimal bacterial growth. The *o*-dihydroxyphenyl groups allow tannins to chelate ferric iron and make it unavailable to bacteria, inhibiting the growth of bacteria such as *E.coli*. However, supplementation of iron can restore the *E. coli* growth, even in the presence of tannins [44]. Tannins bind iron through -dihydroxyphenyl moieties and can form lattices with iron ions, leading to coprecipitation of iron and phenolics from a solution [45]. Moreover, tannins belong to a proposed “four and more metal binding sites” group [46], indicating that they have one of the highest iron binding capacities among phenolic compounds. Iron deprivation leads to a decrease in activity of metalloenzymes and to inhibition of oxidative phosphorylation. Iron complexing may explain the antibacterial activity of tannins (MIC 512 µg/mL), i.e., it may inhibit efflux pumps through interaction of pump proteins that require iron as a cofactor. Their activity against bacteria includes the ability to damage the cell membrane, inhibit enzyme activities and to interact with proteins [47].

At the bacterial cell wall level, tannins inactivate the enzymes involved in cell wall synthesis or bind directly to the peptide glycan layer of the cell wall. Interaction of tannins with the cell wall synthesis makes bacteria more susceptible to osmotic lysis. Tannins also alter the structure of the bacterial membranes in *S. aureus,* which in turn increase fluidity and disrupt formation of virulent membrane vesicles, thus acting as potential enhancers of antibiotics [48] by affecting the membrane potential and increasing membrane permeability. Hexa- and hepta-galloylglucopyranoses form hydrogen bonds with bacterial membrane protein of *S. tphymurium* and *B. cereus*, leading to alteration in the permeability of the cell membrane and to cell death [14]. Procyanidins alter integrity and permeability of the cell wall and membrane by increasing alkaline phosphatase and Na^+^/K^+^-ATPase as well as intracellular Ca^2+^ concentration, which may interfere with protein synthesis and bind to grooves in DNA [49]. Proanthocyanidins can bind lipopolysaccharides of Gram-negative bacteria, leading to destabilization of the integrity of the outer membrane. They act on the outer membrane of *E. coli* and *P. aeruginosa*, penetrating the cell membrane. Tannins have inhibitory effect on numerous bacterial enzymes, such as protease, phospholipase, urease, neuraminidase, and collagenase. Tannic acid reduces the biofilm formation of *P. mirabilis*, inhibiting the expression and activity of the urease gene. Other tannins inhibit collagenase and protease activity. Paeonianin and terchebin have a bacterial inhibitory effect on neuraminidase enzymes secreted by *V. cholerae*, *P. aeruginosa*, and *S. pneumoniae* with IC_50_ values of 15 and 31 μM, respectively [14].

### 2.6. Potential Mechanisms of Actions

Phenolics interact with a broad range of compounds synthesized in biological organisms; these compounds may act as molecular targets for the biological activity of phenolics. Current databases indicate that phenolic compounds such as rutin, quercetin, diosmin, hesperitin and genistein, may interact with a large number of compounds, many of which are involved in microbial/bacterial metabolism [50]. Polyphenols act against bacteria on different levels and metabolic pathways. The structure and formation of nucleic acids can be affected, as can the structures of the cell membrane and of various enzymes (Figure 1).

Polyphenols can directly bind to and damage bacterial cell membranes, even though bacteria have complex multilayered structure that can protect them against environmental factors. The complexity of the cell membrane serves to not only help the bacteria survive, but also to transfer nutrients or waste products. Gram-positive bacteria are the most sensitive to phenolic compounds because of the peptidoglycans on the surface of this type of bacteria and because they lack an outer membrane. The presence of functional groups from the membrane structure cause different reactions to occur at this level. For instance, polyphenols can damage the bacterial cell wall through the hydroxylic groups that bind the peptidoglycans from the membrane structure. The resistance of bacteria to this type of antimicrobial activity depends on both the polyphenol’s structure and the type of bacteria. In contrast to Gram-positive bacteria, the cell membrane on Gram-negative bacteria consist of three principal layers: the outer membrane, the peptidoglycan layer and the inner membrane, all of which are more resistant to the antibacterial activity of phenolic compounds. This resistance is related to the high levels of phospholipids on the lipophilic outer membrane [51,52]. In this case, the antibacterial mechanism involves the accumulation of hydroxylic groups in lipid bilayers, which damage the interaction of lipoproteins and increase the permeability of cell membrane. Polyphenols may also destroy membrane integrity, change cell morphology, affect cell metabolism and leak cellular content. Destruction of the phospholipid bilayer causes cell death through alterations of cell division and physiological functions [20].

Phenolic compounds also have an impact on bacteria DNA synthesis and regulation. The structure consists of hydroxylic groups and an aromatic ring that allows polyphenols to interact with amino or carboxylic groups from proteins [53]. The influence on cell membrane permeability means that polyphenols can lead to leakage of cellular contents, including DNA. Phenolic compounds can also bind to genomic DNA, resulting in changes in the secondary structure and morphology of DNA [54]. Polyphenols such as gallic acid, (-)-epigallocatechin, (-)-epigallocatechin gallate, (+)-catechin, quercetin, hydroxytyrosol, delphinidin and rosmarinic acid produce hydrogen peroxide, which accelerates the self-oxidation of polyphenols. Thus, polyphenols undergo a transformation into pro-oxidants and induce antimicrobial activity, possibly through synthesis of hydrogen peroxide in a concentration dependent manner. Polyphenols alone cannot disrupt the integrity of bacterial DNA. The reaction with hydrogen peroxide changes the expression of bacterial proteins, leading to DNA damage and abnormal transcription [55].

Moreover, phenolic compounds act as antibacterials by hindering enzymatic activity. The mechanism of regulation of such expression is due to the protein-phenolic interactions that occur through covalent or noncovalent interactions and depend on the structural properties of proteins, such as hydrophobicity, molecular weight, conformational configuration, amino acid composition and sequence. These compounds have a metal ion complexation ability, resulting in ligands with iron, cooper and zinc that affect bacterial enzyme activity [53]. Polyphenols change the bacterial metabolism through inhibition of bacterial enzymes such as hydrolase, oxidoreductase, lyase and transfer enzymes. For example, epigallocatechin gallate exhibits antibacterial activity against *S. maltophilia* because of the inhibition of its dihydrofolate reductase [56]. Polyphenoloxidase catalyzes the phenolic compounds’ oxidation, leading to oxidized compounds that inactivate glucan synthase, which is a plasma membrane-bound enzyme. Oxidized polyphenols irreversibly change the glucan synthase structure through covalent modifications. Moreover, some polyphenols such as catechin, epicatechin, pyrogallol and tannic acid inactivate glucan synthase activity even in the absence of polyphenoloxidase [27].

Another mechanism for enzyme inhibition involves the reactions between polyphenols and SH- groups of proteins or non-specific interactions. The degree of inhibition depends on the number of OH groups, and thus, highly oxidized polyphenols have a greater toxicity on microorganisms [57]. Phenolic compounds bind enzymes such as glucosyltransferase B and C, which are responsible for the production of a plaque and contribute to adherence of oral bacteria to the dental surface [58]. The mechanism of enzyme inhibition can occur in combination with damage to the cytoplasmatic membrane. Microorganisms’ membranes contain enzyme systems, and a change in the membrane’s lipids can influence enzyme activity [59].

Another potential target of antibacterial mechanism is the inhibition of gene expression, sometimes related to virulence factors produced by pathogenic bacteria. For instance, flavone and flavonol, respectively, inhibit production of staphyloxantin, a virulence factor produced by *S. aureus*, and gene expression of α-hemolysin and listeriolysin O, a virulence factor produced by *L. monocytogenes*, which causes gastroenteritis and meningitis [60]. Methyl gallate, a phenolic compound, controls gene expression of *C violaceum* and *P. aeruginosa* and shows regulation of autoinducer synthase, lasI and rhlI, and cognate receptors, lasR and rhlR [61]. Phenolic acids such as gallic, protocatechuic and vanillic acids, can downregulate expression of genes responsible for synthesis and the assembly of the Type III Secretion System (T3SS), responsible for invasion of host cells, specifically hilA, hilD, invH, prgH and prgK, as well as virulence such as fliC, which increases motility and sipA, which is responsible for intracellular survival [62].

## 3. Pharmacodynamics

Although phenolics exhibit significant effects on microbial metabolism in in vitro setups, their effectiveness is altered by either passage through the gastrointestinal (GI) tract or by specific environmental conditions. The antibacterial activity of phenolics in humans depends on their bioavailability, and thus on their chemical structure. The functional groups determine the rate of absorption. After intake, some polyphenolic compounds are absorbed at the stomach level, but most of them reach the small intestine. Through the GI tract, various phenolics are subject to possible release of aglycons under either enzyme or microbial hydrolysis. Glycosylated polyphenols are substrated for intestinal enzymes such as lactase phlorizin hydrolase (LPH), cytosolic β-glucosidase (CBG) and catechol-O-methyltransferase (COMT). At the level of epithelial cells of the small intestine, LPH catalyzes the hydrolysis of glycosylated phenolics, leading to increased lipophilicity of compounds and allowing the metabolites to enter the intestine cells through passive diffusion. Inside the cells, phenolic glycosides, which are not substrates for LPH, are hydrolyzed by CBG. Polyphenols that are absorbed reach the liver, where metabolization occurs under the action of hepatic enzymes that catalyze methylation, glucuronidation and sulfation of phenolic compounds [63,64]. These structural transformations in the liver affect biological properties of phenolics prior to the passage in the blood stream. The compounds that reach the cells and tissue level and have antibacterial activity are different from the original compounds [65]. Polyphenols that are unabsorbed in the stomach or small intestine pass into the large intestine. At this level, they are degradated by colonic microflora into phenolic acids or other smaller molecules [66,67].

The bioavailability of phenolic compounds is influenced by different factors related to food, phenolic structure, host and others. Their structure properties cause polyphenols to interact with dietary fiber, starch, fat, dietary proteins, metallic ions and other phenolic structure. The high number of hydroxyl groups allows polyphenols to bind proteins so that they can interact with digestive enzymes such as amylase, lipase and proteases [68]. As a result, their antimicrobial efficiency differs, and improvement of bioavailability and absorption should be sought through chemical functionalization or encapsulation [64].

## 4. Chemical Functionalization and Derivatization of Natural Phenolics for Enhanced Antimicrobial Activity

The antibacterial activity of polyphenols depends on their chemical structure. An augmented activity can be obtained by the addition of new functional groups, hydrolysis or metal complexation, leading to enhanced interactions with different bacterial cell targets (Figure 2).

Among phenolic acids, a series of esters shows modified activity against several bacterial strains of the *Bacillales* family. For instance, against *E. coli* DMF 7503, methyl esters of vanillic, ferulic and caffeic acids, as well as ethyl esters of ferulic and caffeic acids, have similar activity with the original acids, with a MIC of 10 mM, while propyl and butyl esters, exhibit decreased 5 mM MIC values. Against *B. cereus,* methyl esters have the same MIC value as with the acids (5 mM for protocatechuic, genistic and vanillic acids and 10 mM for ferulic acid). The antibacterial activity of ethyl esters increases to a MIC value of 2.50 mM in the case of genistic and ferulic acids. The antibacterial activity against *L. monocytogenes* DMF 5776 has some MIC value for protocatechuic, genistic, vanillic acids and their methyl and ethyl esters. The value of MIC for ferulic and caffeic acid decrease from 10 mM for acids to 5 mM for methyl and ethyl esters. The alkyl esters do not improve the antibacterial activity/mechanism against bacteria, but the introduction of alkyl groups reduces the polarity of the final molecules. The decrease of polarity leads to an increased solubility of phenolic compounds in oils and thus, facilitates the access in lipophilic cell wall of microorganisms, enhancing antibacterial effect [69]. The propyl and butyl esters have very low MIC value (2.5 or less than 1.25 mM), but the increasing of the alkyl chain length increases the toxicity of the phenolic compounds. Obtaining phenolic alkyl esters can occur through the reaction of phenolic compound with alcohols, further solvation and crystallization.

Chitosan, a macromolecular polymer with antioxidant and antimicrobial activity, can be copolymerized with phenolic acids such as gallic, caffeic, genistic or sinapic acids, leading to enhanced activity due to the hydroxyl groups introduced with phenolic acids. Phenolic acids form a complex with chitosan, containing positively charged amino groups, which interacts with the negatively charged bacterial cell membrane and leads to leakage of intracellular components. Depending on the conjugated phenolic acid, these complexes show different antibacterial activity against *B. subtilis, S. aureus, P. aeruginosa* and *E. coli* [70].

Different reactions at the flavonoid skeleton level lead to augmented bacterial inhibition against bacteria, such as methicillin resistant *S. aureus* [71]. The hydroxylation of C5, C7, C3′ and C4′ and geranylation or prenylation at C6 improve the antibacterial effect through different mechanisms. The presence of hydroxyl group in 5, 7 of the ring A and 4′ of the ring B positions as well as the substitution of 3-O-acyl chains with C8 and C10 in ring C increase the antibacterial action against methicillin resistant *S. aureus* of flavonoids belonging to the flavan-3-ol class. The 4′ hydroxyl group is important for the *E. coli* DNA gyrase inhibition. On the other hand, the lipophilic functional groups, such as OCH_3_ at C3′ and C5, lead to a decrease in the antibacterial effect. Moreover, it was observed that the substitution of OCH_3_ at either C6 or C8 increased antibacterial activity of the resultant flavonoids [72].

The solubility of phenolic compounds in biological systems can be also improved through phosphorylation. Some phosphorylated flavonoids, such as quercetin 4′,5-diphosphate, quercetin 3′,4′,3,5,7-pentaphosphate and apigenin 4′,5,7-triphosphate were obtained [73], and their antibacterial activity was measured against *L. monocytogenes*, *A. hydrophila* and *P. aeruginosa.* Against Gram-positive *L. monocytogenes,* the antibacterial activity was enhanced, with the MIC value decreasing from 0.35 mg/mL for quercetin to 0.20 mg/mL for the diphosphated derivative [74]. The synthesis method involves treatment of quercetin with dibenzyl phosphite, and the phosphorylated compound is concentrated and purified through column chromatography.

Moreover, the addition of the sulfonate group increases solubility by increasing acidity [75]. In the case of quercetin 5′-sulfonic acid, the addition of a sulfonate improves the antibacterial activity of quercetin against *L. monocytogenes*, with a decrease in MIC value from 0.35 mg/mL for quercetin to 0.25 mg/mL for the sulfonated compound [74]. The synthesis of quercetin-5′-sulfonic acid uses quercetin and concentrated sulfuric acid [75].

Enzymatic conjugation of flavonoids may enhance antibacterial activity. For example, naringin, a widely distributed flavonoid in plants, can be conjugated with diastase, leading to improved antibacterial activity against *S. aureus*, with an inhibition diameter increased from 0.3–0.56 mm to 0.86 mm for the enzyme conjugated compound. This increase was associated with increased solubility, enhanced cellular uptake, reduced metabolism and better binding to cell components [76].

Among the lignans, conversion of hydroxyl groups to amino groups can increase the antibacterial activity, for example against *B. subtilis, S. aureus* and *S. choleraesuis.* Among functionalization steps, a demethylation is necessary to obtain 9-O,9′-O-demethyl (+)-virgatusin because the methyl groups decrease the antibacterial effect. Then, 9, 9′-hydroxy groups are replaced by carboxylic acid or amino groups. Against *B. subtilis,* the deaminated compound has the same MIC value, but against *S. aureus* and the MIC value decreases from 50 mM to 25 mM and 12.5 mM, respectively. The MIC value for inhibition of Gram-negative bacteria *S. choleraesuis* decreases from 50 mM to 25 mM. In the case of the di-carboxylated compound, the MIC value against either the Gram-negative or Gram-positive strains increases. These MIC values suggest that the either basicity or nucleophilicity lead to higher antibacterial activity [77].

The antibacterial activity of stilbenes is low compared to the other polyphenols and is due to the hydroxyl groups, but methoxylated groups decrease antibacterial effect. Improved antibacterial effect can be obtained by derivatization, for instance, the addition of fluorine substituents to stilbene increases its permeability into the membrane and leads to a changed partition coefficient of the compounds in the bacteria cell [78]. The combinations of stilbene with different antibiotics are used against many bacterial strains, but some strains are resistant to their activity. For example, a dimer stilbene that is effective against drug resistant *S. aureus*, with the new compound having stronger antibacterial activity than resveratrol or other stilbenes, even at low concentrations, could be obtained. Against *S. aureus*, dimer stilbene inhibits growth, antibiotics should complement the treatment to enhance its effect [79].

In the case of tannins, hydrolysis enhances the antibacterial activity. Tannic acid, a natural polymerized molecule that acts against bacteria through iron deprivation, has activity that is dependent on the presence of galloyl groups. Tannic acid is a polygalloyl glucopyranose, and their hydrolysis forms gallic acid and other galloyl groups from unhydrolyzed tannic acid. These compounds can act synergistically, enhancing antibacterial activity through binding to and precipitation of proteins (including enzymes), due to the hydrophobic interactions by gallic acid with surface proteins on bacteria cells [80]. Among condensed tannins, the antibacterial effect can be enhanced through hydrolysis, while grafting tannins onto lignocellulosic materials improves the activity against *S. aureus* and *E. coli.* The polyphenol immobilization into polymer surface involves a phenol-oxidizing enzyme such as laccase. The treatment with laccase is associated with a better retention of tannins via covalent bonding and the antibacterial effect of oligomeric polyphenols is greater because of the higher radical stability. This enzymatic functionalization of tannins shows improved effect against both Gram-positive and Gram-negative bacteria [81].

## 5. Conclusions

Polyphenols, a widespread class of nutraceuticals, have many beneficial effects for human health, especially bacteriostatic or bactericidal effect against both Gram-positive and Gram-negative bacteria. The multitude of phenolic compounds with different structures is a rich source of potential drugs for inhibiting bacterial strains growth. Furthermore, the presence of functional groups makes them suitable to act at the different levels on bacteria cells, leading to damages to cell membranes, enzyme inhibition and DNA intercalation. An overview of their structure and the relationship with the mechanism at different bacterial cell targets provide a new perspective to obtain compounds with enhanced antibacterial activity. Some structural changes, such as the introduction of new functional groups or conjugation, could lead to improved antibacterial effect, making them valuable compounds for human health.

## Figures and Tables

**Figure 1 molecules-28-01114-f001:**
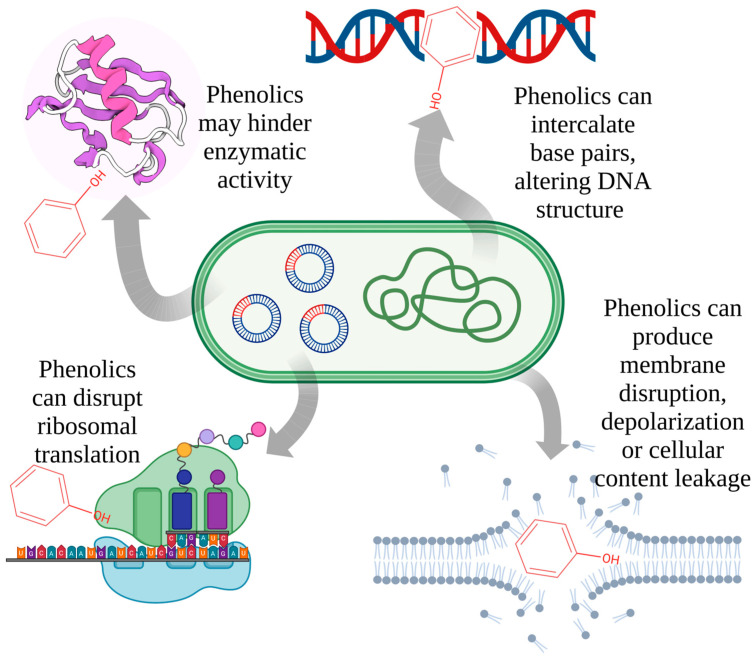
Potential mechanisms of phenolic-bacterial cell interactions (the phenol structure is a placeholder representing various phenolic compounds).

**Figure 2 molecules-28-01114-f002:**
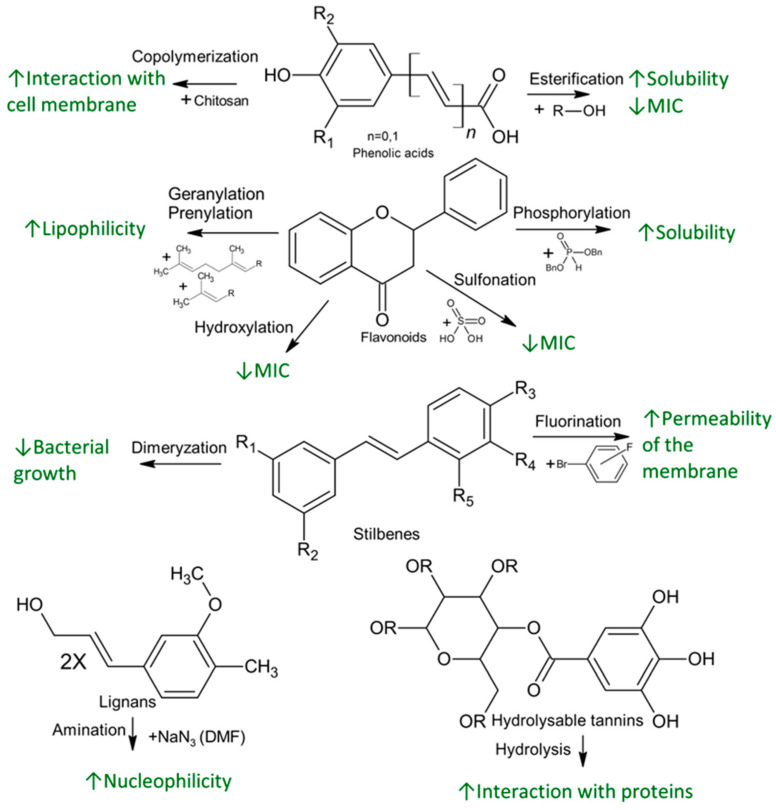
Possible routes for functionalization and increase of activity of phenolic compounds.

**Table 1 molecules-28-01114-t001:** Classification of natural phenolic compounds.

Polyphenols	General Structure	Representative Compounds	Antibacterial Activity
**Phenolic acids**	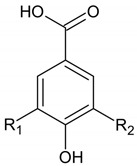	R_1_=H, R_2_=H: 4-Hydroxy Benzoic Acid. R_1_=H, R_2_=OCH_3_: Vanillic Acid. R_1_=OCH_3_, R_2_=OCH_3_: Syringic Acid. R_1_=H, R_2_=OH: Protocatechuic Acid.	Antibacterial effect increases significantly with pH values [10].
a. Benzoic acid derivatives			
b. Derivates of cinnamic acid	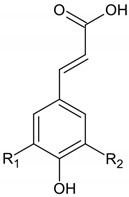	R_1_=H, R_2_=H: *p*-Coumaric Acid. R_1_=H, R_2_=OCH_3_: Ferulic Acid. R_1_=OCH_3_, R_2_=OCH_3_:Sinapic Acid. R_1_=H, R_2_=OH: Caffeic Acid.	
**Flavonoids**	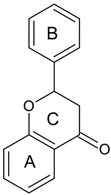	Flavones Flavonols Isoflavones Flavanones Anthocyanidins (flavylium salt) Flavanols	Flavonoids act against bacteria such as *S. aureus and P. aeruginosa* with a very low minimum inhibitory concentration (MIC) value (0.062 µg/mL) [11].
**Lignans**	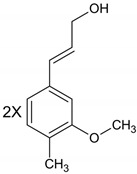	Dibenzylbutanes Dibenzylbutyrolactones, arylnaphthalenes Terahydrofurans Furofurans Dibenzocyclo-octadienes	Due to structural properties, antibacterial activity of lignans is influenced by the stereochemistry of molecules [12].
**Stilbenes**	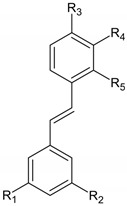	R_1_,R_2_,R_3_=OH, R_4_,R_5_=H: Resveratrol. R_1_,R_2_=OCH_3_, R_3_=OH, R_4_, R_5_=H: Pterostilbene. R_1_,R_2_,R_3_,R_4_=OH, R_5_=H: Piceatannol. R_1_,R_2_,R_3_,R_5_=OH, R_4_=H: Oxyresveratrol.	In combination with antibiotics, some stilbenes can be useful in treating infections caused by multidrug-resistance bacteria [13].
**Tannins**	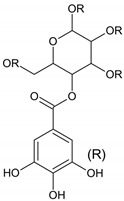	Gallotannins Ellagitannins Complex tannins (Acutissimin A and Eugenigrandin A) Condensed tannins (Procyanidin B_2_, Proanthocyanidin A_1_, Proanthocyanidin A_2_) Low molecular mass phenolics (gallic acid).	Tannin compounds act against bacteria, causing disintegration of bacterial colonies, by interfering with the bacterial cell wall and inhibiting fatty acid biosynthesis pathways [14].
a. Hydrolysable tannins			
b. Nonhydrolysable tannins (Condensed tannins)	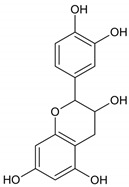		
c. Pseudotannins	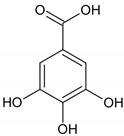		

## Data Availability

Not applicable.

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
