# Peer review of "Future Antimicrobials: Natural and Functionalized Phenolics"

_molecules, 2023, doi:10.3390/molecules28031114_

Round 1

Reviewer 1 Report

The area of this manuscript is very interesting and relevant, as it is really important to clear out in much more details about the Future Antimicrobials: Natural and Functionalized Phenolics. The purpose of the work is clearly stated. The research was logically planned and the discription was properly selected. This includes the Classification of polyphenols, Biosynthesis, natural and synthetic sources, Antibacterial activity, Pharmacodynamics and Chemical functionalisation and derivatisation of natural phenolics for enhanced antimicrobial activity. So, I honestly welcome such studies and the present work includes a lot of good reviews.

1.Line 17, The current review summarizes some of the most important methods of functionalization of natural phenolic compounds and the effects on their antimicrobial activity, the word of methods may be not accurate.

2. In the Abstract part. The information about the background is too long, and the author should give some important information in this part.

3. Classification of natural phenolic compounds, but the author showed the Antibacterial activity in this part, while showed the antibacterial activity in another part.

4. in line 98, Potential mechanisms of actions, but the author respectively showed the antibacterial activity of different compounds, maybe the potential mechanisms of actions should be at the end of this part.

5. If the author could provide the supplement levels about different compounds on antibacterial activity, it would be better.

6. in line 116, there should be a space between peptidoglycan.

7. in line 364. the first time you mention GI, you should give the full name, please check the full text.

8. in line 376, there is a grammar error in this sentence “The antibacterial activity of polyphenols depends on their chemical structure and augmented activity can be obtained by addition of new functional groups”.

9. Discussion part needs improvement, more up to date citation should be added.

Author Response

We thank Reviewer for helpful suggestions. Replies are given below.

The area of this manuscript is very interesting and relevant, as it is really important to clear out in much more details about the Future Antimicrobials: Natural and Functionalized Phenolics. The purpose of the work is clearly stated. The research was logically planned and the discription was properly selected. This includes the Classification of polyphenols, Biosynthesis, natural and synthetic sources, Antibacterial activity, Pharmacodynamics and Chemical functionalisation and derivatisation of natural phenolics for enhanced antimicrobial activity. So, I honestly welcome such studies and the present work includes a lot of good reviews.

1.Line 17, The current review summarizes some of the most important methods of functionalization of natural phenolic compounds and the effects on their antimicrobial activity, the word of methods may be not accurate.

The word methods was replaced.

  1. In the Abstract part. The information about the background is too long, and the author should give some important information in this part.

The abstract was revised.

  1. Classification of natural phenolic compounds, but the author showed the Antibacterial activity in this part, while showed the antibacterial activity in another part.

The title of the subsection was modified, to reflect the classification related to antibacterial activity of phenolics.

  1. in line 98, Potential mechanisms of actions, but the author respectively showed the antibacterial activity of different compounds, maybe the potential mechanisms of actions should be at the end of this part.

The first subsection was moved to the last part of the chapter.

  1. If the author could provide the supplement levels about different compounds on antibacterial activity, it would be better.

MIC values were introduced.

  1. in line 116, there should be a space between peptidoglycan.

The word peptidoglycan was replaced throughout the document with peptido glycan.

  1. in line 364. the first time you mention GI, you should give the full name, please check the full text.

GI was defined.

  1. in line 376, there is a grammar error in this sentence “The antibacterial activity of polyphenols depends on their chemical structure and augmented activity can be obtained by addition of new functional groups”.

The sentence was rephrased.

  1. Discussion part needs improvement, more up to date citation should be added.

Old citations were replaced with up to date citations.

Reviewer 2 Report

The manuscript titled"Future antimicrobials: natural and functionalized phenolics" demonstrated the activity of phenolic compounds as antibacterial.

Kindly,

1- All comments are involved in the attached pdf

2- I recommend a schematic diagram for the structure activity relationship.

Thank you

Reviewer 3 Report

The manuscript under consideration is a review concerning the antimicrobial activities of different phenolic compounds, natural or chemically functionalised. After a brief introduction on the chemical classes which comprise the wide family of natural phenols and polyphenols, their sources and functions, the paper revolves mainly around the different mode of actions against several microorganisms discussed on the base of the natural phenolic class. A concise discourse about functionalised phenols concludes the paper.

The review covered a highly appealing topic, which could be of interest for several researchers, from chemists to biologists due to the relevance of the subject. References came from mostly recent publications and no self-citations were detected. However, the discussion of the topic, even if extensive, could not be considered complete, the references provided are not comprehensive and the manuscript is not clear and well-organized requiring extensive editing of both English language and style. In particular:

1.     Page 1, Vasile Stoleru is missing from the indication on how to cite the manuscript (left column of the page).

2.     The name of the microorganisms must be reported in italic and have to be consistent through all the paper (e.g., C. albicans or Candida albicans, not both).

3.     Bibliography is not uniform and should be standardised.

4.     Chemical names must be reported in the correct form (e.g., p- and not p-, o- and not O-).

5.     A number of major typographical (e.g., missing space between the last author and the conjunction and, the conjunction and reported twice) and grammatical mistakes which made the understandability of the manuscript difficult were noted that should be corrected. Moreover, British and American spelling have been used interchangeably (for example hydrolysable and non-hydrolyzable in Table 1). This could be made more consistent in favour of one.

6.     The images of the chemical structures reported in Table 1 are not consistent and of poor quality. An appropriate editor is recommended to draw chemical structures. Furthermore, additional structures in the text would be useful for a better comprehension of the concepts (e.g., line 237 ring B of flavonoids which is not reported in the text).

7.     Colloquial terms as “doesn’t” (line 273) “can’t” (line 256) “don’t” (line 389) should be avoided.

8.     There is no uniformity in the text in the indication of common names (e.g., in Table 1 tannins reported with capital letters and without in the same context, as also vancomycin-resistant without capital letter and Methicillin-resistant with capital letter in the same sentence). Moreover, in Table 1 some names are reported incorrectly (for example pi-ceatannol instead of piceatannol, as also acu-tissimin A and Euge-nigrandin A).  

9.     Several repetitions through the manuscript made the reading difficult and redundant. Sometimes the concepts are summarily reported, whereas in other cases futile details are listed.

10.  Numerous statements are not adequately supported by previous literature, resulting unclear and in some cases misleading (some examples are lines 128, 134, 158, 188, 281, 319, 332).

11.  The paragraph about chemical functionalisation and derivatisation could be an added value to the manuscript but the presence of experimental protocols is not appropriate with the aim of a review.

Author Response

We thank Reviewer for helpful suggestions. Replies are given below.

The manuscript under consideration is a review concerning the antimicrobial activities of different phenolic compounds, natural or chemically functionalised. After a brief introduction on the chemical classes which comprise the wide family of natural phenols and polyphenols, their sources and functions, the paper revolves mainly around the different mode of actions against several microorganisms discussed on the base of the natural phenolic class. A concise discourse about functionalised phenols concludes the paper.

The review covered a highly appealing topic, which could be of interest for several researchers, from chemists to biologists due to the relevance of the subject. References came from mostly recent publications and no self-citations were detected. However, the discussion of the topic, even if extensive, could not be considered complete, the references provided are not comprehensive and the manuscript is not clear and well-organized requiring extensive editing of both English language and style. In particular:

  1. Page 1, Vasile Stoleru is missing from the indication on how to cite the manuscript (left column of the page).

The name was added.

  1. The name of the microorganisms must be reported in italic and have to be consistent through all the paper (e.g., albicans or Candida albicans, not both).

The name of microorganisms was modified.

  1. Bibliography is not uniform and should be standardised.

Bibliography was revised

  1. Chemical names must be reported in the correct form (e.g., p- and not p-, o- and not O-).

Chemical names were modified.

  1. A number of major typographical (e.g., missing space between the last author and the conjunction and, the conjunction and reported twice) and grammatical mistakes which made the understandability of the manuscript difficult were noted that should be corrected. Moreover, British and American spelling have been used interchangeably (for example hydrolysable and non-hydrolyzable in Table 1). This could be made more consistent in favour of one.

The text was revised

  1. The images of the chemical structures reported in Table 1 are not consistent and of poor quality. An appropriate editor is recommended to draw chemical structures. Furthermore, additional structures in the text would be useful for a better comprehension of the concepts (e.g., line 237 ring B of flavonoids which is not reported in the text).

The images were improved.

  1. Colloquial terms as “doesn’t” (line 273) “can’t” (line 256) “don’t” (line 389) should be avoided.

The terms were modified.

  1. There is no uniformity in the text in the indication of common names (e.g., in Table 1 tannins reported with capital letters and without in the same context, as also vancomycin-resistant without capital letter and Methicillin-resistant with capital letter in the same sentence). Moreover, in Table 1 some names are reported incorrectly (for example pi-ceatannol instead of piceatannol, as also acu-tissimin A and Euge-nigrandin A).

The text was revised.

  1. Several repetitions through the manuscript made the reading difficult and redundant. Sometimes the concepts are summarily reported, whereas in other cases futile details are listed.

The meaning of text was revised

  1. Numerous statements are not adequately supported by previous literature, resulting unclear and in some cases misleading (some examples are lines 128, 134, 158, 188, 281, 319, 332).

      The indicated paragraphs were revised.

  1. The paragraph about chemical functionalisation and derivatisation could be an added value to the manuscript but the presence of experimental protocols is not appropriate with the aim of a review.

The paragraph on functionalisation was revised

Round 2

Reviewer 2 Report

accept in the current from.

Thank you

Reviewer 3 Report

The revised version of the paper "Future antimicrobials: natural and functionalized phenolics" benefits from a structural reorganization which enhance the comprehensibility of the review. The figure and the scheme included in the new version assist the reader in the understanding of the topic. In particular, the scheme regarding the functionalization of phenolic compounds results clear and well-written. Some minor typos are still present. For example: British English and American English, some chemical names and microorganisms name not in italic (e.g., lines 248 and 261), missing space between letters and brackets (e.g. line 277). Considering the relevance of the topic I suggest accepting the work after minor revisions.